# Aroma Profile and Chemical Composition of Reverse Osmosis and Nanofiltration Concentrates of Red Wine Cabernet Sauvignon

**DOI:** 10.3390/molecules26040874

**Published:** 2021-02-07

**Authors:** Ivana Ivić, Mirela Kopjar, Vladimir Jukić, Martina Bošnjak, Matea Maglica, Josip Mesić, Anita Pichler

**Affiliations:** 1Faculty of Food Technology Osijek, Josip Juraj Strossmayer University, F. Kuhača 18, 31000 Osijek, Croatia; iivic@ptfos.hr (I.I.); mirela.kopjar@ptfos.hr (M.K.); mbosnjak@ptfos.hr (M.B.); mmaglica@ptfos.hr (M.M.); 2Faculty of Agrobiotechnical Sciences Osijek, Josip Juraj Strossmayer University, V. Preloga 1, 31000 Osijek, Croatia; vladimir.jukic@fazos.hr; 3Polytechnic in Požega, Vukovarska 17, 34000 Požega, Croatia; jmesic@vup.hr

**Keywords:** Cabernet Sauvignon red wine, aroma compounds, reverse osmosis, nanofiltration, retention, chemical composition

## Abstract

Wine aroma represents one of the main properties that determines the consumer acceptance of the wine. It is different for each wine variety and depends on a large number of various chemical compounds. The aim of this study was to prepare red wine concentrates with enriched aroma compounds and chemical composition. For that purpose, Cabernet Sauvignon red wine variety was concentrated by reverse osmosis (RO) and nanofiltration (NF) processes under different operating conditions. Different pressures (2.5, 3.5, 4.5 and 5.5 MPa) and temperature regimes (with and without cooling) were applied on Alfa Laval LabUnit M20 equipped with six composite polyamide RO98pHt M20 or NF M20 membranes. Higher pressure increased the retention of sugars, SO_2_, total and volatile acids and ethanol, but the temperature increment had opposite effect. Both membranes were permeable for water, ethanol, acetic acid, 4-ethylphenol and 4-ethylguaiacol and their concentration decreased after wine filtration. RO98pHt membranes retained higher concentrations of total aroma compounds than NF membranes, but both processes, reverse osmosis and nanofiltration, resulted in retentates with different aroma profiles comparing to the initial wine. The retention of individual compounds depended on several factors (chemical structure, stability, polarity, applied processing parameters, etc.).

## 1. Introduction

Wine is a complex alcoholic drink, where the main components are water and ethanol, followed by sugars, acids, higher alcohols, aroma and phenolic compounds, etc. The quality of wine will depend on all mentioned parameters. Ethanol is produced during alcoholic fermentation and its content is usually up to 15 vol.%, depending on grape variety and vinification techniques. Higher concentrations of ethanol contribute to sweet taste and burning sensation [1]. Level of reducing sugar mostly depends on fermentation conditions and grape juice chemical properties, and it greatly affects wine taste. Wines are classified from dry to sweet according to the reducing sugar content [2]. Most common organic acids found in wine are tartaric, malic, lactic, citric, succinic and other [3].

Aroma represents one of the most important quality parameter of wine. It is a result of a combination of several hundred various compounds responsible for the taste and odour of the wine. Compounds that are volatile at room temperature are responsible for wine scent [4] and they can be divided on primary, secondary and tertiary aroma. Primary aroma originates from grape berries and it is distinctive for each variety, but it also depends on environmental factors, climate, berries conditions, etc. The amount of aroma compounds that will be transferred into the grape must and wine depends on vinification techniques, especially maceration and extraction [2,5]. Secondary aroma compounds are formed during alcoholic (converting sugar to ethanol) and lactic (converting malic to lactic acid) fermentation. Ethanol and glycerol are quantitatively dominating alcohols. Most common carboxylic acids that contribute to wine aroma are acetic acid, propanoic acid, butanoic acid, decanoic acid, etc. Acetic acid is formed in larger amounts comparing to the rest of volatile acids and contributes to the wine aroma unless it is formed in higher concentrations (above 0.9 g/L), as a result of wine spoilage [5,6,7]. The type and concentration of each compound depends on several factors: chemical composition of grape must, yeast strain, fermentation temperature and other vinification techniques [8]. Tertiary aroma (“bouquet”) is formed during wine storage and aging. The final aroma depends on chemical composition of wine, pH, containers for wine aging (wood, stainless steel), maturation strategies, temperature, aeration, etc. [9].

It is visible that wine chemical composition and aroma profile depends on numerous factors that can sometimes result in a product that does not meet the standards (lower concentrations of desirable compounds or higher concentrations of compounds that negatively affect wine quality). In those cases, when wine aroma enhancement or correction is needed, membrane filtration by reverse osmosis (RO) or nanofiltration (NF) could be applied. Membrane filtration is a pressure-driven, low energy, high efficiency operation that is conducted at mild temperatures [10]. They are based on selective membrane application that split the initial feed on retentate that retains on the membrane, and permeate that yields through it [11]. Membrane properties are usually expressed through molecular weight cut-off (MWCO) that for nanofiltration membranes usually vary between 200 and 1000 Da, and for reverse osmosis they do not exceed 200 Da, depending on the manufacturer [12]. This means that RO and NF membranes permeate mostly molecules and ions with molecular weight lower than MWCO value (1 Da is usually equalized with 1 g/mol), that makes them applicable for liquid concentration. The molecules and ions that are retained on the membrane create osmotic pressure and concentration polarization effect on the surface. This requires the use of high pressures (up to 60 MPa or higher) to establish a permeate flux [13].

Described membrane characteristic makes them applicable for wine concentration. Water, ethanol, acetic acid and several low molecular weight compounds pass through the membrane, but high percentage of valuable bioactive compounds is retained on it. Therefore, RO and NF processes can be used for wine dealcoholisation with minimized organoleptic changes [14,15,16,17], for grape juice or wine concentration in order to increase or correct the content of sugars, polyphenols and aroma compounds [17,18,19,20,21], for acetic acid correction [22,23,24] or for eliminating “bad” aroma compounds [25,26,27].

The aim of this study was to obtain wine concentrates with enriched and corrected chemical composition and aroma profile, and to examine the influence of different membrane type and operating conditions on final product. For that purpose, membranes for reverse osmosis and nanofiltration process were used, applying four pressures (2.5, 3.5, 4.5 and 5.5 MPa) and two temperature regimes (with and without cooling) for both processes. During concentration, permeate flux and retentate temperature were measured. In obtained samples, sugars, acids, sulphur dioxide and ethanol were determined. Aroma compounds were analysed by gas chromatograph with mass spectrometer in obtained retentates and initial wine. In our previous study Cabernet Sauvignon was subjected to process of reverse osmosis and nanofiltration at different pressures and temperature regimes to determine the influence of RO and NF membranes on phenolics and colour compounds retention [28].

## 2. Results

### 2.1. Processing Parameters

The concentration of Cabernet Sauvignon red wine variety by reverse osmosis (RO) and nanofiltration (NF) was conducted at four different transmembrane pressures (2.5, 3.5, 4.5 and 5.5 MPa) and two temperature regimes (with and without cooling). The initial temperature of wine in all experiments was 20 °C. The pressure increase resulted in higher retentate temperature through shorter time (Figure 1), especially when cooling was not applied. The highest temperature was measured at 5.5 MPa without cooling during both processes, reverse osmosis (56 °C) and nanofiltration (47 °C).

The reverse osmosis process lasted longer and resulted in higher retentate temperatures, comparing to the nanofiltration process at the same pressures. Further, the increase of working pressure increased the retentate temperature and permeate flux (Figure 2). Higher temperatures resulted in lower viscosity of the feed, that leads to higher permeate flux [18,29]. The highest average permeate flux during nanofiltration (39.45 L/m^2^h) and reverse osmosis process (17.75 L/m^2^h) was measured at 5.5 MPa without cooling. Decreasing the pressure and cooling the retentate resulted in lower permeate flux and longer process duration.

### 2.2. Chemical Composition of Initial Wine and Retentates

Reducing sugars, free and total SO_2_, total and volatile acids, alcohol content and total extract in initial Cabernet Sauvignon wine and retentates obtained by reverse osmosis and nanofiltration are presented in Table 1 and Table 2, respectively.

The results showed that reverse osmosis and nanofiltration processes resulted in higher reducing sugars content in retentates than in initial wine. Application of RO membranes resulted in slightly higher concentrations of sugars than the application of NF membranes. Pressure increase and cooling regime were more favourable for sugars retention in NF retentates than lower pressure (2.5 and 3.5 MPa) and higher temperatures. However, RO membranes retained same amount of sugar at 4.5 and 5.5 MPa with and without cooling (no significant difference among concentrations). Similar trend was observed for total acids retention: higher temperatures and lower pressure resulted in lower retention. RO membranes retained slightly higher concentrations of total acids than NF ones at same operating conditions. Volatile acids content decreased during both processes comparing to the initial content in wine (1.44 g/L). The decrease of volatile acids was a result of membrane permeability to acetic acid that is the volatile acids representative. The highest concentrations of volatile acids were retained in RO retentates at 5.5 MPa with cooling (1.40 g/L).

Alcohol content (mostly ethanol) in retentates followed the same trend as volatile acids. Retentates contained lower content of alcohol than initial wine (13.62 vol.%), with highest concentrations at 5.5 MPa with cooling (11.01 vol.% in RO retentate and 10.38 vol.% in NF retentate). NF membranes showed slightly higher permeability to ethanol. The lowest concentrations of free and total SO_2_ were found in RO and NF retentates at 2.5 MPa at both temperature regimes. Slightly higher retention of SO_2_ was observed with pressure increase and retentate cooling, as with RO membranes application.

### 2.3. Aroma Compounds Retention

Aroma compounds identified in initial Cabernet Sauvignon red wine and retentates obtained by reverse osmosis and nanofiltration, and their retention indices and odour descriptions are shown in Table 3.

For better display, all 47 volatile compounds were divided into six groups (acids, alcohols, carbonyl compounds, terpenes, esters and volatile phenols). These compounds are characteristic of red wine, especially Cabernet Sauvignon wine variety, as shown in previous studies [30,31]. Concentrations of individual compounds in initial wine and obtained retentates are presented in Table 4 and Table 5.

Seven acids (acetic, octanoic, nonanoic, decanoic, lauric, myristic and palmitic acid) were identified in initial wine and retentates, where the acetic acid had the highest concentration (682.5 μg/L) in initial wine. After membrane filtration treatment, all obtained retentates contained significantly lower amounts of acetic acid. The lowest concentration among RO retentates was evaluated at 2.5 MPa without cooling, meaning that the decrease of pressure and increase of temperature lowered the acetic acid retention. The increase of transmembrane pressure resulted in higher retention of acetic acid, and the highest concentration were evaluated at 4.5 MPa (428.7 μg/L) and 5.5 MPa (436.4 μg/L) with cooling in RO retentates. NF retentates contained lower concentrations of acetic acid than RO retentates, with highest one obtained at 5.5 MPa with cooling, 365.8 µg/L.

The rest of identified acids had higher concentrations in the retentate than in the initial wine, but the processing parameters did not affect each compound the same way. Octanoic, decanoic and myristic acid concentrations followed the above-mentioned trend: higher pressure leads to higher retention, with slight decrease when cooling was not applied. Their concentrations (in initial wine were 25.6, 65.4 and 24.8 µg/L, respectively) increased during wine concentration by reverse osmosis and nanofiltration, especially at 5.5 MPa with cooling where the highest concentrations were found. Pressure increase resulted in higher retention of lauric acid at both processes, but concentrations were slightly higher at regime without cooling in comparison to the one with cooling. Reverse osmosis was more favourable for palmitic acid concentration comparing to the nanofiltration process. The initial concentration in wine (14.0 µg/L), in NF retentates decreased by 50% or more (the lowest concentration was found in NF retentate at 2.5 and 3.5 MPa without cooling, 5.2 and 5.3 µg/L, respectively). The highest concentration of palmitic acid was found in RO retentate at 5.5 MPa with cooling (38.9 µg/L). Nonanoic acid was identified only in RO retentates and it was not detected in initial wine and NF retentates. It is possible that the concentration of nonanoic acid in initial wine was under the threshold of detection for the applied GC/MS method, but its concentration increased during reverse osmosis process. The content of nonanoic acid increased with the higher pressure (the highest concentration was found in RO retentate at 5.5 MPa with cooling, 16.0 μg/L). Slight decrease of concentration of this acid was noticed when cooling was not applied comparing to the regime with cooling. In Table 4 and Table 5 the sum of concentrations for each group of volatiles (acids, alcohols, carbonyl compounds, terpenes, esters and phenols) was calculated. According to the total sum of all acids in samples, a decrease of acids concentrations was observed in retentates comparing to the initial wine (828.8 µg/L), except for RO retentates obtained at 4.5 and 5.5 MPa with cooling and 5.5 MPa without cooling. Lower concentrations of total acids after RO and NF wine treatment were a result of membranes permeability to acetic acid, which had the largest share among acids (82.3% in initial wine).

Among nine identified higher alcohols (isoamyl alcohol, 1-butanol, 2,3-butanediol, 1-hexanol, methionol, benzyl alcohol, 1-octanol, 2-phenylethanol and dodecanol), in initial wine isoamyl alcohol (3.98 mg/L), 2-phenylethanol (1.86 mg/L) and 1-butanol (1.06 mg/L) had the highest concentrations. The concentration of 2,3-butanediol was 0.30 mg/L and the concentrations of the rest of alcohols were below 0.05 mg/L in initial wine. An increase of these concentrations was observed in retentates obtained by reverse osmosis and nanofiltration with cooling, especially at 4.5 and 5.5 MPa. However, regime without cooling resulted in a significant loss in higher alcohol content, with the lowest concentrations measured at 2.5 MPa. In the RO retentate at 2.5 MPa without cooling, 1-butanol and 1-hexanol were not detected. Benzyl alcohol was not detected in RO retentates obtained at the regime without cooling regardless the applied pressure, and it was not detected in any NF retentate. Methionol was found in each RO retentate, with the highest concentration at 5.5 MPa with cooling (32.3 µg/L). However, only two NF retentate (at 2.5 and 3.5 MPa with cooling) contained methionol and the concentrations were lower than in initial wine, 20.6 µg/L. Nanofiltration membranes showed lower ability to retain higher alcohols comparing to the reverse osmosis ones according to the total amount of alcohols. This was mostly visible when cooling was applied. Without cooling regime and higher temperatures resulted in a significant loss of alcohols during both processes.

Regarding the influence of applied pressure and temperature on retention, carbonyl compounds (4-propylbenzaldehyde, geranyl acetone, lily aldehyde and hexyl cinnamaldehyde) and terpenes (α-terpinolene, nerol, β-citronellol, β-damascenone and phenanthrene) concentrations followed a similar trend as higher alcohols. The highest concentrations of these compounds were found in retentates obtained by reverse osmosis at 4.5 and 5.5 MPa with cooling. Significant loss of most carbonyl compounds and terpenes was observed with the regime without cooling comparing to the regime with cooling. The highest increase was noticed for 4-propylbenzaldehyde whose concentration increased from 6.7 μg/L in initial wine to 124.9 μg/L in the RO retentate at 5.5 MPa with cooling, and only 25.5 µg/L in the NF retentate at 5.5 MPa with cooling. In NF retentates, absence of cooling had less impact on several carbonyl compounds and terpenes retention than transmembrane pressure. The retention of 4-propylbenzaldehyde, lily aldehyde, hexyl cinnamaldehyde and β-damascenone was higher when cooling was not applied in the NF retentates than in the ones obtained with cooling. Temperature increase had low influence on the concentrations of lilly aldehyde and β-damascenone during RO process at 4.5 and 5.5 MPa (there was no significant difference between two temperature regimes at those pressures). The total concentrations of carbonyl compounds (20.9 µg/L) and terpenes (69.4 µg/L) in initial wine increased during RO and NF processes. RO membranes retained higher concentrations of carbonyl compounds and terpenes than initial wine, with the highest total concentrations obtained at 5.5 MPa with cooling (173.2 and 211.5 µg/L, respectively).

Esters were the largest group with 19 identified compounds. In initial wine, diethyl succinate had the highest concentration (0.73 mg/L), followed by ethyl octanoate (210.7 μg/L), ethyl hydrogen succinate (183.0 μg/L), ethyl palmitate (107.5 µg/L) and diisobutyl phthalate (103.4 μg/L). The reverse osmosis process with cooling resulted in an increase of concentrations of most esters, where the retention was greater at higher pressures (4.5 and 5.5 MPa). The increase of temperature resulted in a decrease of ester content, but the increase of pressure at regime without cooling had different effect on individual esters. The concentrations of ethyl 4-hydroxybutanoate, ethyl octanoate, ethyl pentadecanoate and ethyl linoleate decreased when higher pressure was applied at regime without cooling. There was no significant difference in concentrations of ethyl vanillate, ethyl laurate and ethyl stearate among RO retentates obtained without cooling at all pressures, while the pressure increment resulted in higher retention of the rest of esters. During nanofiltration process, the retention of esters was also greater at higher pressures (4.5 and 5.5 MPa) and lower temperatures (regime with cooling). The concentrations of ethyl 4-hydroxybutanoate, ethyl pentadecanoate, ethyl oleate and ethyl stearate decreased as the pressure increased when cooling was not applied. The content of ethyl octanoate, methyl palmitate and ethyl linoleate did not significantly change (*p* < 0.05) during pressure increase in NF retentates obtained without cooling. Comparing two types of membranes, RO membranes showed higher ability to retain esters than NF ones. The highest total concentration of esters was found in RO retentate obtained at 5.5 MPa with cooling (4.31 mg/L). At those operating conditions during NF process, the total concentration of esters was 2.59 mg/L, that was still higher than the concentration in initial wine (1.83 mg/L).

Table 4 and Table 5 also present the content of volatile phenols. One of them, 2,4-Di-T-butylphenol, had a high concentration in initial wine, 1.11 mg/L, and it increased with the pressure increment during membrane filtration. At both temperature regimes at 2.5 MPa, nanofiltration retentates contained slightly higher concentrations of this phenol than reverse osmosis ones. Higher pressures were more favourable for reverse osmosis and the highest concentrations were measured at 5.5 MPa with cooling (1.71 mg/L). Temperature increase (without cooling processes) resulted in lower concentration of 2,4-Di-T-butylphenol in RO and NF retentates, comparing to the ones obtained with cooling. For 4-ethylphenol and 4-ethylguaiacol, higher pressure and lower temperature resulted also in higher retention in RO and NF retentates. The concentrations of 4-ethylphenol and 4-ethylguaiacol in the initial wine were 624.8 and 20.9 µg/L, respectively, but after concentration processes their content was significantly lower, with the exception of RO retentate at 5.5 MPa with cooling, where the retention of 4-ethylphenol was high (647.2 µg/L). Both processes resulted in removing 4-ethylguaiacol from retentates, except at 5.5 MPa with cooling where 14.3 and 12.9 µg/L was found in RO and NF retentates, respectively.

In this study, all aroma compounds identified in initial wine and retentates obtained by reverse osmosis and nanofiltration processes at different operating conditions were divided according to their main flavour note. Odour descriptions have been presented in Table 3. There were eight groups of different flavour notes: fatty, green, floral, citrus, fruity, smoky, faint and other. The last group included acetic acid (vinegar aroma), 1-butanol (fusel oil aroma), methionol (sulphurous note), ethyl 4-hydroxybutanoate (caramellic aroma) and ethyl pentadecanoate (honey aroma). For each sample, total sum of concentrations in each group was calculated and principal component analysis was made. Principal component analysis (PCA) is a multivariate statistical analysis method for representing complex and large dataset with reduced dimensionality, increased interpretability and minimized data loss [32]. In Figure 3, it can be observed that the principal component 1 (PC1), accounting 76.76% of total variance, separated the samples according to the operating condition used during filtration (membrane type, pressure and temperature regime). All NF retentates (except the one obtained at 5.5 MPa with cooling), RO retentates at 2.5 and 3.5 MPa without cooling and initial wine are located on the negative side of PC1. Principal component 2 (PC2; 12.07% of total variance) contributed to the separation of samples according to the dominating flavour note, from citrus to smoky. On the positive side of PC2, all NF retentate could be found. Initial wine and RO retentates are located on negative side of PC2, except for RO retentates obtained at 4.5 MPa at both temperature regimes. It can be observed that NF retentates had similar aroma profile that differed from the RO retentates and initial wine flavour composition. Slight difference can be observed in NF retentates at 4.5 and 5.5 MPa with cooling with more intense citrus note than the rest of NF retentates. Aroma profile of RO retentates depended on applied pressure and temperature. RO retentate obtained at 2.5 MPa without cooling had similar aroma profile as initial wine and they were clustered at the negative sides of both principal components. Retentates obtained by reverse osmosis at 4.5 and 5.5 MPa without cooling differed from the one obtained with cooling or at lower pressures.

## 3. Discussion

Membrane filtration processes are often used for aroma, colour, sugar or ethanol correction in wine. They are pressure-driven operations that are influenced by transmembrane pressure, temperature, operating time, membrane type, number of used membranes and module arrangement, MWCO, velocity of the feed, membrane fouling, concentration polarization and osmotic pressure that occur on the membrane surface [19,33]. The higher the pressure, the sooner the membrane fouling and concentration polarization occur, leading to decrease of permeate flux and higher retention of most compounds [13,34]. Further, pressure increase leads to higher interaction of water with hydrophilic active layer of the membrane, resulting in higher permeability of water than other compounds [22]. On the other hand, temperature increment (without cooling regime) resulted in a loss of bioactive compounds in retentates due to thermal degradation or decreased viscosity of the feed at higher temperatures [18,29]. Rezzadori et al. [35] stated that permeate flux during filtration of *n*-hexane solution was several times higher than water flux because *n*-hexane has three times lower viscosity than water.

Reverse osmosis and nanofiltration membranes were permeable for ethanol and its content decreased during red wine concentration in all retentates comparing to the initial wine. Concentrates obtained in this study could be used for production of low alcohol wine. Besides health, there are several reasons for lowering or removing ethanol from wine, like social impacts and alcohol abuse or financial impacts and jurisdiction taxes for excessive alcohol content in wine [36]. Reverse osmosis and nanofiltration proved to be more effective for ethanol removal than thermal processes, such as traditional evaporation process, that results in high thermal degradation of wine components and aroma loss [37]. On the other hand, RO and NF membranes retained sugars and their concentration increased during both processes comparing to the initial wine. This ability could be used for sugar correction in wine or grape juice [14,25]. Higher pressure and lower temperature resulted in higher retention of most compounds (sugar, SO_2_, acids and ethanol), meaning that optimal operating conditions will depend on desired final product.

Different transmembrane pressures (2.5, 3.5, 4.5 and 5.5 MPa) and temperature regimes (with and without cooling) did not have the same effect on all volatile compounds. Besides processing parameters, the retention of individual aroma compound depended also on membrane characteristics, molecular weight, volatility and activity coefficient of each compound [38] and on the wine non-volatile matrix [17]. Reverse osmosis membranes have greater ability to retain low molecular weight compounds due to their small pore size and MWCO value. This can cause higher energy consumption and severe membrane fouling than nanofiltration process [18,39]. However, the results in this study showed that the reverse osmosis and nanofiltration membranes are permeable for acetic acid due to low molecular weight of this compound (60.05 g/mol). Acetic acid is the main component in wine volatile acids group. It is produced during alcoholic or lactic fermentation as secondary product. In small amounts it contributes the wine aroma, but higher concentrations lead to bitter and sour aftertaste or vinegar-like aroma and then it is considered as wine fault [6]. Therefore, the ability of RO and NF membranes to permeate acetic acid could be used for acetic acid correction in wine. The pH of wine (especially pH 3.2) usually favours the acetic acid permeability [40] and around 60% of the initial concentration of acetic acid can permeate through the membrane depending on the processing conditions [25]. This is the reason why total volatile acids content (Table 1 and Table 2) was lower in all obtained retentates than in initial wine. The retention of acetic acid decreases with lower transmembrane pressure and higher temperature [25,40]. It has been reported that acetic acid can be separated from other organic matrices (monosaccharide or xylose) using RO and NF membranes [23,24,41].

The content of rest of carboxylic acids in retentates increased during wine concentration, meaning that the RO98pHt and NF M20 membranes are not permeable for higher acids. Their retention depended on acid chemical properties and operating conditions. Slight loss of these compounds occurred when cooling was not applied and at lower pressures. Reverse osmosis membranes showed greater ability to retain acids than nanofiltration ones.

Higher alcohols at concentrations below 300 mg/L contribute to the desirable wine aroma, but if their concentration exceeds 400 mg/L, a negative effect was observed [30]. In this study, during reverse osmosis and nanofiltration processes, higher pressures were suitable for higher alcohol retention, but the temperature increase (without cooling regime) resulted in significantly lower retention of alcohols, depending on their vapour pressure and volatility [17]. In global, higher pressure and lower temperature resulted also in higher retention of carbonyl compounds, terpenes and esters, but operating conditions did not affect each compounds equally. López et al. [42] stated that the rejection of a compound depends on volatility of the compound and polarity of the membrane. The polyamide membranes showed higher retention of volatile compounds than cellulose acetate membranes. The retention of individual aroma compounds in that study were explained through polarity and hydrophobicity of the membranes. Stronger polar membrane would attract polar organic compounds on the surface and the permeability would be higher. On the other hand, hydrophobic membranes showed greater rejection for high polar organic compounds. For example, hexanol has a hydrophobic character and it is attracted to the hydrophobic part of membrane, which increases its permeability, especially with higher temperatures. This is consistent with our finding, because its concentration in retentates was not significantly higher than in initial wine, and during processes without cooling, its concentration was significantly reduced. Further, a notable loss of several esters was observed during both membrane processes used in this study, such as ethyl hexanoate, ethyl octanoate or ethyl decanoate. A previous study [43] stated that loss of certain esters during wine dealcoholisation could occur mostly due to their hydrophobicity. However, the retention of aroma compounds depends on several other factors, such as wine non-volatile matrix as mentioned before. Polyphenols represent major non-volatile component in red wine and they interact with the aroma compounds through hydrogen bonding, increasing their stability [17,44].

Regarding volatile phenols, three compounds were monitored. The highest concentration among them was measured for 2,4-Di-T-butylphenol and its concentration increased during both processes, reverse osmosis and nanofiltration. However, increased temperature at without cooling regime resulted in a slight loss of 2,4-Di-T-butylphenol, but its retention increased at higher pressures. Further, 4-ethylphenol and 4-ethylguaiacol are low molecular weight compounds (122.2 g/mol for 4-ethylphenol and 152.2 g/mol for 4-ethylguaiacol) and they permeate through the membranes. The NF membranes showed slightly higher permeability for both compounds than reverse osmosis ones, as expected. Their retention at both processes increased with higher pressure and lower temperature, but their concentrations were visible lower than in the initial wine, and in most retentates 4-ethylguaiacol was completely removed. This is a desirable behaviour, because 4-ethylphenol and 4-ethylguaiacol are usually associated with unpleasant aroma of smoke, stable, horse sweat or medicinal aids at higher concentrations [45]. They are products of *Brettanomyces* yeast metabolism that becomes active after fermentation, mostly during wine storage in wooden barrels or due to inadequate hygiene in wineries. These phenols are mostly associated to red wines and are formed by the decarboxylation and *Brettanomyces* mediated reduction of the corresponding grape-derived hydroxycinnamic acids (*p*-coumaric and ferulic acid). Higher concentrations (several hundred µg/L for 4-ethylphenol and 50 to 100 µg/L for 4-ethylguaiacol) negatively affect wine aroma and are considered as wine spoilage [27]. It is reported [26] that reverse osmosis can effectively remove or lower the concentration of these compounds due to their low molecular weight. Reverse osmosis process was used to split the permeate containing 4-ethylphenol and 4-ethylguaiacol from the initial feed. Such obtained permeate is subjected to hydrophobic adsorbent resin treatment in order to remove mentioned compounds and then returned into the retentate. Fudge et al. [46] used similar method to remove smoke-derived volatiles, such as guaiacol and 4-methylguaiacol.

Therefore, reverse osmosis and nanofiltration can be applied for red wine concentration in order to enhance desirable wine components or to remove compounds that have a negative effect on its sensory properties. Different operating conditions did not affect each compound equally that resulted in different aroma profiles among retentates, according to the PCA analysis. Nanofiltration retentates had very similar aroma profile, while the reverse osmosis ones were significantly different regarding applied pressure and temperature regime. Although reverse osmosis membranes proved to be better in terms of total volatiles retention, the application of nanofiltration is preferred due to higher permeate flux, shorter process duration and lower final retentate temperatures at same pressures than reverse osmosis [37]. Increased temperature causes higher evaporation and thermal degradation of volatiles and other wine components and this is the main reason why cooling should be applied. At both processes, reverse osmosis and nanofiltration, the retention of total volatiles was significantly higher when the retentate temperature was lower. Similar results have been obtained during concentration of grape must [21] and chokeberry juice [33,38], although juice or grape must have high soluble solids level and the membrane efficiency and working pressure have smaller effect on their concentration comparing to the concentration of wine with lower soluble solids content.

## 4. Materials and Methods

### 4.1. Reagents and Standards

Copper(II) sulphate pentahydrate, citric acid monohydrate, anhydrous sodium carbonate, potassium hexacyanoferrate(II) trihydrate, zinc acetate, calcium carbonate, potassium iodide, sulphuric acid, potassium thiocyanate, starch, phenolphthalein, sodium hydroxide, sodium thiosulfate and sodium chloride were purchased from Kemika (Zagreb, Croatia). Myrtenol standard was obtained from Sigma Aldrich (St. Lois, MO, USA) and C7-C30 saturated alkanes standard from Merck (Darmstadt, Germany). Helium for gas chromatography was obtained from Messer, Austria.

### 4.2. Wine

Cabernet Sauvignon red wine variety (vintage 2018) was produced at Faculty of Agrobiotechnical Sciences, cultivation area Mandićevac, Đakovo vineyard, Croatia.

### 4.3. Preparation of Cabernet Sauvignon Red Wine Concentrates

Concentration of Cabernet Sauvignon red wine was conducted on a laboratory plate-and-frame filter, LabUnit M20 (De Danske Sukkerfabrikker, Nakskov, Denmark). Four different pressures (2.5, 3.5, 4.5 and 5.5 MPa) and two temperature regimes (with and without cooling) were applied to obtain the samples. The membrane module was equipped with six composite Alfa Laval RO98pHt M20 (for reverse osmosis processes) or six composite Alfa Laval NF M20 flat sheet polyamide membrane (for nanofiltration processes). The main characteristics of RO98pHt membranes and NF membranes are presented in Table 6.

Filtration surface of a membrane was 0.0289 m^2^. The initial volume of wine for each experimental run was 3 L and the initial wine temperature was 20 °C. During concentration, permeate volume and retentate temperature was measured every 4 min. At the end of each process, the permeate volume was 1.7 L and retentate volume was 1.3 L.

### 4.4. Chemical Composition Analysis

Reducing sugars in wine and retentates were determined according to Luff-Schoorl method. Free and total SO_2_ were measured by titration with iodine and starch as indicator. Titration with 0.25 mol/L NaOH was applied for total acids measurement and 0.1 mol/L NaOH for volatile acids determination with phenolphthalein indicator. Total acids were expressed as g/L of tartaric acid and volatile acids as g/L of acetic acid [47]. Alcohol content and total extract were measured on Electronic hydrostatic balance Super Alcomat (Gibertini Elettronic, Milano, Italy) and digital distilling unit Super Dee (Gibertini Elettronic, Milano, Italy). All results were expressed as average value of three repetitions.

### 4.5. Aroma Analysis

Aroma profiles of all samples were determined on Agilent 7890B gas chromatograph equipped with Agilent 5977A mass spectrometer (Agilent Technologies, Santa Clara, CA, USA). Solid-phase microextraction (SPME) was used as an extraction method. In 10 mL glass headspace vial 5 mL of sample was mixed with 1 g of NaCl. For quantification of aroma compounds 5 μL of internal standard (myrtenol in concentration of 1 mg/L) was injected in each sample. Prepared vials with samples were heated on magnetic stirrer at 40 °C at 300 rpm. The adsorption time on the SPME fibre coated with polydimethylsiloxane/divinylbenzene (PDMS/DVB) sorbent (65 μm, Supelco, Bellefonte, PA, USA) was 45 min. Further, the SPME fibre was transferred into GC injector port where the volatiles were desorbed at 250 °C for 7 min, splitless mode. In the GC oven, HP-5MS (30 m × 0.25 mm × 0.25 μm) column was installed and following method was applied: oven was heated from 40 °C (held for 10 min) to 120 °C at 3 °C/min, then to 250 °C at 10 °C/min, using helium (He) 5.0 (purity 99.999%) as carrier gas (1 mL/min). The temperatures of MS Source and MS Quad were set at 230 °C and 150 °C, respectively. Mass range (*m*/*z*) was from 40 to 400 and the ionization energy 70 eV. Obtained peaks and mass spectra were compared with NIST (National Institute of Standards and Technology, Gaithersburg, MD, USA) and Wiley mass spectral database. For each compound, linear retention index was calculated [48], using a C7–C30 saturated alkanes standards analysed under same GC/MS conditions. Samples were analysed in triplicates and the results were expressed as average value. The limit of detection (LOD), limit of quantitation (LOQ) and mass to charge ratio (*m*/*z*) for monitored compounds are presented in Table 7. Single compound quantitation was obtained by single ion monitoring (SIM) mode. The LOD and LOQ represent the lowest amount of an analyte in a sample that can be detected or quantified, respectively [49,50]. They have been estimated using the instrumental signal to noise ratio (SNR) of 3:1 for LOD and 10:1 for LOQ. Signal to noise ratio was calculated by Agilent’s MassHunter software.

### 4.6. Statistical Analysis

Average value and standard deviation were calculated for each result. Statistical analyses of results were carried out with STATISTICA 13.1 (StatSoft Inc., Tulsa, OK, USA) software program, where the analysis of variance (ANOVA) and Fisher’s least significant difference (LSD) test (*p* < 0.05) were applied. For principal component analysis (PCA) of wine and retentates aroma profile, all volatile compounds were divided into eight main groups according to their odour description (fatty, green, floral, citrus, fruity, smoky, faint and other).

## 5. Conclusions

Reverse osmosis and nanofiltration processes proved to be suitable for wine chemical composition and aroma enhancement or correction. The concentrations of sugar, total SO_2_, acids and extract increased after RO and NF wine treatment. Higher retention of mentioned compounds were achieved at higher pressure and cooling regime. Both membrane types differently affected the retention of individual aroma compounds. Their retention depended on chemical properties of each component and initial feed, operating conditions and membranes characteristics. In global, RO98pHt M20 membranes were slightly better in terms of aroma compounds retention than NF M20 membranes, although nanofiltration process was shorter and resulted in higher permeate flux and lower final retentate temperature than RO process. Low molecular weight compounds, such as ethanol, acetic acid, 4-ethylphenol and 4-ethylguaiacol can be removed by reverse osmosis and nanofiltration process due to their permeability through membrane. Lower pressures and higher temperatures were more favourable for these operations, but this also results in higher loss of valuable volatiles, meaning that optimal processing conditions should be applied in order to obtain desirable properties of red wine concentrates.

## Figures and Tables

**Figure 1 molecules-26-00874-f001:**
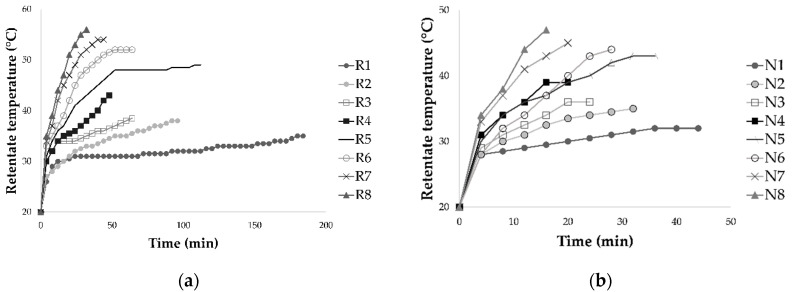
Influence of process duration (min) on retentate temperature (°C) during concentration of Cabernet Sauvignon red wine by reverse osmosis (**a**) and nanofiltration (**b**). Abbreviations: R—reverse osmosis retentate; N—nanofiltration retentate; 1–2.5 MPa with cooling; 2–3.5 MPa with cooling; 3–4.5 MPa with cooling; 4–5.5 MPa with cooling; 5–2.5 MPa without cooling; 6–3.5 MPa without cooling; 7–4.5 MPa without cooling; 8–5.5 MPa without cooling.

**Figure 2 molecules-26-00874-f002:**
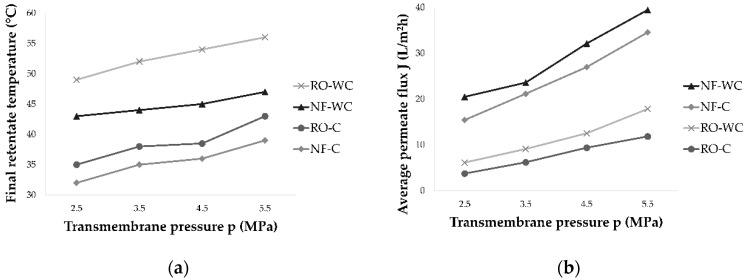
Influence of transmembrane pressure (2.5, 3.5, 4.5 and 5.5 MPa) on: (**a**) final retentate temperature (°C) and (**b**) average permeate flux (L/m^2^h) during reverse osmosis (RO) and nanofiltration (NF) of Cabernet Sauvignon red wine with (C) and without cooling (WC).

**Figure 3 molecules-26-00874-f003:**
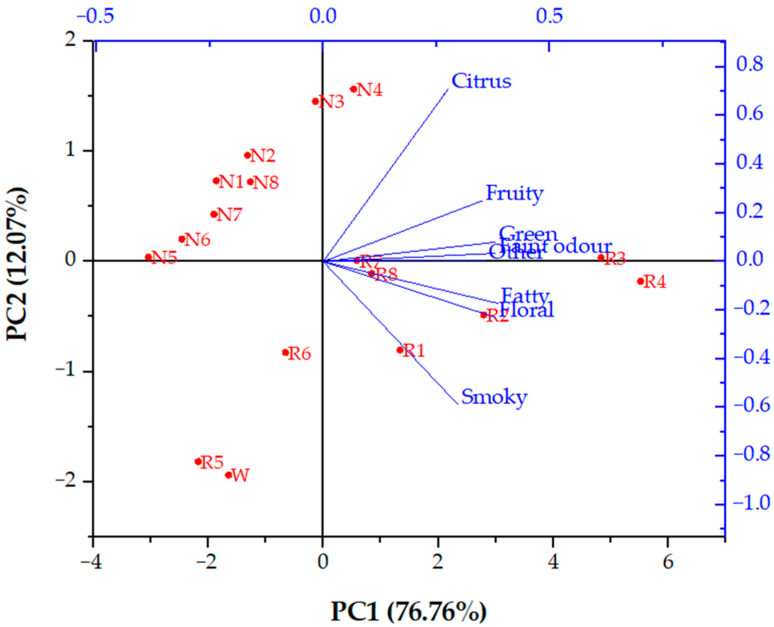
Principal component analysis (PCA) biplot of aroma profiles of initial wine (W) and retentates obtained by reverse osmosis (R) and nanofiltration (N). Abbreviations: 1–2.5 MPa with cooling; 2–3.5 MPa with cooling; 3–4.5 MPa with cooling; 4–5.5 MPa with cooling; 5–2.5 MPa without cooling; 6–3.5 MPa without cooling; 7–4.5 MPa without cooling; 8–5.5 MPa without cooling.

**Table 1 molecules-26-00874-t001:** Chemical composition of initial Cabernet Sauvignon wine variety and reverse osmosis retentates at 2.5, 3.5, 4.5 and 5.5 MPa with cooling (C) and without cooling (WC).

Sample	Reducing Sugars (g/L)	Free SO_2_ (mg/L)	Total SO_2_ (mg/L)	Total Acids (g/L)	Volatile Acids (g/L)	Alcohol (vol.%)
W	3.66 ± 0.2 ^a^	40.1 ± 0.1 ^e^	64.0 ± 0.1 ^a^	6.56 ± 0.01 ^a^	1.44 ± 0.00 ^d^	13.62 ± 0.00 ^g^
R1	6.59 ± 0.1 ^c^	35.3 ± 0.3 ^b^	70.4 ± 0.1 ^b^	7.50 ± 0.01 ^c^	1.33 ± 0.02 ^b^	9.34 ± 0.00 ^c^
R2	6.79 ± 0.2 ^c^	38.4 ± 0.1 ^d^	72.5 ± 0.3 ^c^	7.81 ± 0.04 ^d^	1.34 ± 0.02 ^b^	9.93 ± 0.00 ^d^
R3	7.17 ± 0.1 ^d^	40.5 ± 0.3 ^e^	74.7 ± 0.3 ^d^	8.13 ± 0.04 ^e^	1.37 ± 0.01 ^b^	10.56 ± 0.00 ^e^
R4	7.19 ± 0.1 ^d^	51.2 ± 0.1 ^g^	76.8 ± 0.1 ^e^	8.44 ± 0.01 ^f^	1.40 ± 0.01 ^c^	11.01 ± 0.00 ^f^
R5	6.18 ± 0.2 ^b^	34.1 ± 0.3 ^a^	70.4 ± 0.1 ^b^	7.19 ± 0.04 ^b^	1.25 ± 0.01 ^a^	7.85 ± 0.00 ^a^
R6	6.58 ± 0.1 ^c^	37.4 ± 0.1 ^c^	72.5 ± 0.3 ^c^	7.50 ± 0.01 ^c^	1.27 ± 0.02 ^a^	9.04 ± 0.00 ^b^
R7	7.16 ± 0.1 ^d^	40.7 ± 0.3 ^e^	74.7 ± 0.3 ^d^	7.81 ± 0.04 ^d^	1.31 ± 0.02 ^b^	9.83 ± 0.00 ^d^
R8	7.17 ± 0.3 ^d^	49.1 ± 0.3 ^f^	74.7 ± 0.3 ^d^	8.13 ± 0.04 ^e^	1.35 ± 0.01 ^b^	10.39 ± 0.00 ^e^

Different superscript letters in the same column represent statistically different values (*p* < 0.05; ANOVA, Fisher’s (LSD) test). Abbreviations: W—initial wine; R—reverse osmosis retentate; 1–2.5 MPa with cooling; 2–3.5 MPa with cooling; 3–4.5 MPa with cooling; 4–5.5 MPa with cooling; 5–2.5 MPa without cooling; 6–3.5 MPa without cooling; 7–4.5 MPa without cooling; 8–5.5 MPa without cooling.

**Table 2 molecules-26-00874-t002:** Chemical composition of initial Cabernet Sauvignon wine variety and nanofiltration retentates at 2.5, 3.5, 4.5 and 5.5 MPa with cooling (C) and without cooling (WC).

Sample	Reducing Sugars (g/L)	Free SO_2_ (mg/L)	Total SO_2_ (mg/L)	Total Acids (g/L)	Volatile Acids (g/L)	Alcohol (vol.%)
W	3.66 ± 0.2 ^a^	40.1 ± 0.1 ^e^	64.0 ± 0.1 ^a^	6.56 ± 0.01 ^c^	1.44 ± 0.00 ^c^	13.62 ± 0.00 ^g^
N1	5.21 ± 0.2 ^b,c^	32.0 ± 0.1 ^a^	68.3 ± 0.3 ^c^	6.25 ± 0.04 ^b^	1.23 ± 0.01 ^a^	9.61 ± 0.00 ^b^
N2	5.64 ± 0.2 ^c^	36.3 ± 0.3 ^c^	72.5 ± 0.3 ^e^	6.56 ± 0.01 ^c^	1.30 ± 0.03 ^b^	9.82 ± 0.00 ^d^
N3	6.86 ± 0.1 ^e^	42.7 ± 0.3 ^f^	74.7 ± 0.3 ^f^	6.88 ± 0.04 ^d^	1.33 ± 0.01 ^b^	10.24 ± 0.00 ^e^
N4	7.02 ± 0.2 ^f^	46.9 ± 0.3 ^g^	74.7 ± 0.3 ^f^	7.50 ± 0.01 ^f^	1.35 ± 0.02 ^b^	10.38 ± 0.00 ^f^
N5	5.03 ± 0.2 ^b^	32.0 ± 0.1 ^a^	66.1 ± 0.1 ^b^	5.94 ± 0.04 ^a^	1.20 ± 0.01 ^a^	9.26 ± 0.00 ^a^
N6	5.33 ± 0.2 ^b,c^	34.1 ± 0.3 ^b^	70.4 ± 0.1 ^d^	6.25 ± 0.04 ^b^	1.27 ± 0.01 ^b^	9.77 ± 0.00 ^c^
N7	6.15 ± 0.2 ^d^	36.3 ± 0.3 ^c^	72.5 ± 0.3 ^e^	6.56 ± 0.01 ^c^	1.33 ± 0.01 ^b^	9.83 ± 0.00 ^d^
N8	6.57 ± 0.3 ^d,e^	38.4 ± 0.1 ^d^	72.5 ± 0.3 ^e^	7.19 ± 0.04 ^e^	1.33 ± 0.01 ^b^	9.74 ± 0.00 ^c^

Different superscript letters in the same column represent statistically different values (*p* < 0.05; ANOVA, Fisher’s (LSD) test). Abbreviations: W—initial wine; N—nanofiltration retentate; 1–2.5 MPa with cooling; 2–3.5 MPa with cooling; 3–4.5 MPa with cooling; 4–5.5 MPa with cooling; 5–2.5 MPa without cooling; 6–3.5 MPa without cooling; 7–4.5 MPa without cooling; 8–5.5 MPa without cooling.

**Table 3 molecules-26-00874-t003:** Linear retention indices (LRI) and odour description of volatile compounds identified in Cabernet Sauvignon wine variety and retentates obtained by reverse osmosis and nanofiltration at 2.5, 3.5, 4.5 and 5.5 MPa with and without cooling.

Compound	LRI	Odour	Compound	LRI	Odour
Acids			Esters		
Acetic acid	622	vinegar	Ethyl hexanoate	997	fruity
Octanoic acid	1199	fatty	Ethyl 4-hydroxybutanoate	1060	caramellic
Nonanoic acid	1265	fatty	Diethyl succinate	1179	fruity
Decanoic acid	1376	fatty	Ethyl octanoate	1191	fruity
Lauric acid	1556	fatty	Ethyl hydrogen succinate	1198	faint
Myristic acid	1749	fatty	Phenethyl acetate	1248	floral
Palmitic acid	2004	fatty	Ethyl decanoate	1391	fruity
Alcohols			Ethyl vanillate	1580	smoky
Isoamyl alcohol	734	fruity	Ethyl laurate	1584	fatty
1-butanol	752	fusel oil	Hexyl salicylate	1667	green
2,3-butanediol	804	fruity	Ethyl myristate	1778	fatty
1-hexanol	868	green	Diisobutyl phthalate	1859	faint
Methionol	981	sulphurous	Ethyl pentadecanoate	1880	honey
Benzyl alcohol	1037	fruity	Methyl palmitate	1907	fatty
1-octanol	1071	green	Dibutyl phthalate	1953	faint
2-phenylethanol	1103	floral	Ethyl palmitate	1978	fatty
Dodecanol	1469	fatty	Ethyl linoleate	2146	fatty
Carbonyl compounds			Ethyl oleate	2152	fatty
4-propylbenzaldehyde	1261	faint	Ethyl stearate	2176	fatty
Geranyl acetone	1448	floral	Volatile phenols		
Lily aldehyde	1517	floral	4-ethylphenol	1166	smoky
Hexyl cinnamaldehyde	1738	floral	4-ethylguaiacol	1268	smoky
Terpenes			2,4-Di-T-butylphenol	1501	faint
α-terpinolene	1092	citrus			
Nerol	1218	citrus			
β-citronellol	1223	citrus			
β-damascenone	1377	fruity			
Phenanthrene	1772	faint			

**Table 4 molecules-26-00874-t004:** Aroma compounds identified in Cabernet Sauvignon red wine and reverse osmosis retentates at 2.5, 3.5, 4.5 and 5.5 MPa with cooling and without cooling.

Compound	W	R1	R2	R3	R4	R5	R6	R7	R8
**∑** **Acids (μg/L)**	828.8 ± 29.9 ^d^	699.9 ± 13.7 ^c^	723.4 ± 11.5 ^c^	820.2 ± 14.8 ^d^	900.2 ± 14.9 ^e^	498.3 ± 8.4 ^a^	581.5 ± 12.2 ^b^	684.3 ± 24.1 ^c^	820.5 ± 14.5 ^d^
Acetic acid (μg/L)	682.5 ± 27.6 ^g^	388.9 ± 2.5 ^d^	396.4 ± 3.0 ^d^	428.7 ± 1.0 ^e^	436.4 ± 0.9 ^f^	242.0 ± 0.2 ^a^	256.9 ± 2.4 ^b^	313.5 ± 11.1 ^c^	426.7 ± 0.5 ^d^
Octanoic acid (μg/L)	25.6 ± 0.3 ^a^	68.6 ± 1.3 ^d^	62.1 ± 0.9 ^c^	66.4 ± 1.0 ^c,d^	79.9 ± 0.9 ^e^	46.5 ± 0.5 ^b^	64.7 ± 0.4 ^c^	72.1 ± 2.3 ^e^	76.2 ± 1.6 ^f^
Nonanoic acid (μg/L)	-	10.8 ± 0.1 ^b^	11.7 ± 0.7 ^c^	12.2 ± 1.1 ^c^	16.0 ± 0.3 ^e^	8.0 ± 0.1 ^a^	8.1 ± 0.1 ^a^	12.4 ± 0.6 ^c^	14.3 ± 0.9 ^d^
Decanoic acid (μg/L)	65.4 ± 0.8 ^a^	138.8 ± 8.0 ^b^	137.4 ± 4.9 ^b^	186.1 ± 10.1 ^d^	206.0 ± 6.3 ^e^	142.5 ± 6.0 ^b^	155.6 ± 7.1 ^b,c^	153.9 ± 5.7 ^b,c^	166.0 ± 10.0 ^c^
Lauric acid (μg/L)	16.5 ± 0.1 ^a^	31.8 ± 0.9 ^b^	36.7 ± 0.4 ^c^	39.2 ± 0.4 ^d^	52.7 ± 3.5 ^f^	29.9 ± 0.8 ^b^	46.4 ± 1.2 ^e^	60.0 ± 1.9 ^g^	65.0 ± 0.9 ^h^
Myristic acid (μg/L)	24.8 ± 0.5 ^b^	40.7 ± 0.5 ^c^	52.2 ± 0.6 ^d^	52.4 ± 1.1 ^d^	70.3 ± 1.3 ^f^	18.4 ± 0.7 ^a^	42.1 ± 0.9 ^c^	63.0 ± 2.3 ^e^	62.6 ± 0.2 ^e^
Palmitic acid (μg/L)	14.0 ± 0.6 ^d^	20.3 ± 0.4 ^e^	26.9 ± 1.0 ^f^	35.2 ± 0.1 ^g^	38.9 ± 1.7 ^h^	11.0 ± 0.1 ^c^	7.7 ± 0.1 ^a^	9.4 ± 0.2 ^b^	9.7 ± 0.4 ^b^
**∑** **Alcohols (mg/L)**	7.29 ± 0.21 ^b^	10.93 ± 0.30 ^c^	14.18 ± 0.40 ^d^	25.18 ± 1.48 ^e^	25.17 ± 0.39 ^e^	5.02 ± 0.11 ^a^	7.96 ± 0.13 ^b^	11.00 ± 0.29 ^c^	11.86 ± 0.69 ^c^
Isoamyl alcohol (mg/L)	3.98 ± 0.12 ^b^	3.88 ± 0.17 ^b^	5.15 ± 0.15 ^c^	14.95 ± 0.47 ^e^	14.43 ± 0.13 ^e^	2.18 ± 0.07 ^a^	4.14 ± 0.03 ^b^	6.94 ± 0.15 ^d^	7.68 ± 0.61 ^d^
1-butanol (mg/L)	1.06 ± 0.06 ^c^	2.61 ± 0.03 ^d^	4.13 ± 0.14 ^e^	4.46 ± 0.07 ^f^	4.38 ± 0.03 ^f^	-	0.42 ± 0.03 ^a^	0.58 ± 0.02 ^b^	0.60 ± 0.02 ^b^
2,3-butanediol (mg/L)	0.30 ± 0.02 ^b^	0.53 ± 0.06 ^c^	0.59 ± 0.03 ^c^	1.06 ± 0.03 ^d^	1.60 ± 0.11 ^e^	0.16 ± 0.01 ^a^	0.54 ± 0.01 ^c^	0.53 ± 0.01 ^c^	0.56 ± 0.01 ^c^
1-hexanol (μg/L)	42.4 ± 0.7 ^c^	46.5 ± 1.2 ^d^	56.0 ± 1.4 ^e^	59.1 ± 2.6 ^e^	68.8 ± 1.9 ^f^	-	0.8 ± 0.1 ^a^	0.6 ± 0.1 ^a^	2.2 ± 0.1 ^b^
Methionol (μg/L)	20.6 ± 0.8 ^d^	25.3 ± 0.9 ^e^	26.3 ± 0.2 ^e^	29.8 ± 0.5 ^f^	32.3 ± 1.0 ^g^	11.9 ± 1.7 ^a^	14.1 ± 0.2 ^b^	15.0 ± 0.9 ^b,c^	16.2 ± 0.2 ^c^
Benzyl alcohol (μg/L)	7.2 ± 0.1 ^a^	17.8 ± 0.1 ^d^	13.2 ± 0.1 ^b^	15.2 ± 0.3 ^c^	14.8 ± 0.7 ^c^	-	-	-	-
1-octanol (μg/L)	14.4 ± 0.3 ^a^	35.7 ± 0.1 ^e^	41.6 ± 0.3 ^f^	42.0 ± 1.4 ^f^	41.8 ± 2.4 ^f^	25.6 ± 0.6 ^b^	28.6 ± 0.1 ^c^	30.2 ± 0.2 ^d^	30.2 ± 0.3 ^d^
2-phenylethanol (mg/L)	1.86 ± 0.01 ^a^	3.77 ± 0.04 ^d^	4.16 ± 0.08 ^e^	4.52 ± 0.09 ^f^	4.53 ± 0.11 ^f^	2.63 ± 0.03 ^b^	2.81 ± 0.06 ^c^	2.89 ± 0.11 ^c^	2.96 ± 0.05 ^c^
Dodecanol (μg/L)	6.0 ± 0.1 ^a^	9.8 ± 0.1 ^c^	15.0 ± 0.6 ^e^	44.5 ± 1.6 ^f^	69.3 ± 2.2 ^g^	8.0 ± 0.2 ^b^	9.0 ± 0.1 ^b^	12.1 ± 1.3 ^d^	14.3 ± 0.2 ^e^
**∑** **Carbonyl compounds (μg/L)**	20.9 ± 0.8 ^a^	37.3 ± 0.6 ^c^	67.6 ± 3.0 ^f^	145.1 ± 6.8 ^g^	173.2 ± 9.9 ^h^	27.4 ± 0.7 ^b^	43.8 ± 1.1 ^d^	56.7 ± 1.9 ^e^	63.8 ± 2.7 ^f^
4-propylbenzaldehyde (μg/L)	6.7 ± 0.3 ^a^	9.9 ± 0.1 ^a^	29.1 ± 1.1 ^c^	99.9 ± 5.7 ^d^	124.9 ± 8.0 ^e^	11.6 ± 0.4 ^a,b^	18.2 ± 0.7 ^b^	26.1 ± 0.6 ^c^	30.9 ± 2.0 ^c^
Geranyl acetone (μg/L)	5.3 ± 0.1 ^a^	11.8 ± 0.1 ^c^	16.6 ± 0.2 ^f^	19.6 ± 0.1 ^g^	22.5 ± 0.8 ^h^	6.2 ± 0.1 ^b^	12.2 ± 0.2 ^c^	13.0 ± 0.6 ^d^	15.0 ± 0.2 ^e^
Lily aldehyde (μg/L)	4.4 ± 0.2 ^a^	6.8 ± 0.1 ^c^	7.0 ± 0.2 ^c^	7.9 ± 0.3 ^d^	7.8 ± 0.3 ^d^	4.9 ± 0.1 ^a^	5.7 ± 0.1 ^b^	8.3 ± 0.4 ^d^	8.3 ± 0.3 ^d^
Hexyl cinnamaldehyde (μg/L)	4.5 ± 0.2 ^a^	8.8 ± 0.3 ^c^	14.9 ± 1.5 ^e^	17.7 ± 0.7 ^f^	18.0 ± 0.8 ^f^	4.7 ± 0.1 ^a^	7.7 ± 0.1 ^b^	9.3 ± 0.3 ^c,d^	9.6 ± 0.2 ^d^
**∑Terpenes (μg/L)**	69.4 ± 3.8 ^a^	140.3 ± 0.9 ^c^	163.0 ± 4.5 ^d^	186.8 ± 1.6 ^e^	211.5 ± 8.4 ^f^	91.8 ± 3.3 ^b^	132.2 ± 5.3 ^c^	170.4 ± 4.2 ^d^	190.5 ± 5.4 ^e^
α-terpinolene (μg/L)	15.9 ± 0.7 ^b^	22.8 ± 0.1 ^c^	32.4 ± 1.4 ^e^	33.8 ± 0.3 ^e^	32.9 ± 1.2 ^e^	10.6 ± 0.3 ^a^	21.5 ± 1.1 ^c^	26.9 ± 1.1 ^d^	26.8 ± 0.2 ^d^
Nerol (μg/L)	5.1 ± 0.1 ^a^	8.6 ± 0.1 ^d^	8.7 ± 0.6 ^d^	9.5 ± 0.2 ^e^	9.7 ± 0.3 ^e^	5.7 ± 0.1 ^a^	6.9 ± 0.1 ^b^	7.9 ± 0.4 ^c^	8.3 ± 0.4 ^c,d^
β-citronellol (μg/L)	24.2 ± 1.0 ^b^	33.3 ± 0.1 ^c^	37.5 ± 1.2 ^d^	42.3 ± 0.5 ^e^	50.2 ± 0.1 ^f^	10.9 ± 1.1 ^a^	26.5 ± 1.8 ^b^	37.3 ± 1.2 ^d^	35.8 ± 1.5 ^d^
β-damascenone (μg/L)	18.9 ± 1.9 ^a^	68.4 ± 0.5 ^b,c^	77.2 ± 1.2 ^c^	93.0 ± 0.5 ^c^	110.1 ± 6.7 ^e^	60.2 ± 1.7 ^b^	72.4 ± 2.2 ^c^	92.7 ± 1.2 ^d^	112.9 ± 3.2 ^e^
Phenanthrene (μg/L)	5.3 ± 0.1 ^c^	7.2 ± 0.1 ^f^	7.2 ± 0.1 ^f^	8.2 ± 0.1 ^g^	8.6 ± 0.1 ^h^	4.4 ± 0.1 ^a^	4.9 ± 0.1 ^b^	5.6 ± 0.3 ^d^	6.7 ± 0.1 ^e^
**∑Esters (mg/L)**	1.83 ± 0.06 ^a^	3.36 ± 0.08 ^d^	3.43 ± 0.08 ^d^	4.08 ± 0.06 ^e^	4.31 ± 0.07 ^f^	2.43 ± 0.04 ^b^	2.62 ± 0.05 ^b^	3.06 ± 0.05 ^c^	3.19 ± 0.07 ^c,d^
Ethyl hexanoate (μg/L)	66.7 ± 2.4 ^d^	57.8 ± 3.8 ^c^	59.7 ± 1.1 ^c^	68.2 ± 2.4 ^d^	70.4 ± 1.6 ^d^	37.8 ± 2.2 ^a^	38.4 ± 0.7 ^a^	47.3 ± 1.9 ^b^	47.6 ± 0.4 ^b^
Ethyl 4-hydroxybutanoate (μg/L)	50.2 ± 0.9 ^b^	55.0 ± 0.5 ^c^	65.0 ± 0.1 ^d^	95.7 ± 1.1 ^e^	106.9 ± 1.6 ^f^	45.5 ± 0.3 ^a^	50.1 ± 1.4 ^b^	47.4 ± 1.8 ^a,b^	48.5 ± 0.5 ^b^
Diethyl succinate (mg/L)	0.73 ± 0.02 ^a^	1.45 ± 0.04 ^c^	1.45 ± 0.03 ^c^	1.75 ± 0.02 ^e^	1.84 ± 0.01 ^f^	1.20 ± 0.01 ^b^	1.29 ± 0.01 ^b^	1.57 ± 0.02 ^d^	1.65 ± 0.04 ^d^
Ethyl octanoate (μg/L)	210.7 ± 14.7 ^b^	244.8 ± 7.3 ^c^	240.7 ± 3.8 ^c^	243.7 ± 2.9 ^c^	242.4 ± 0.1 ^c^	117.5 ± 0.1 ^a^	116.5 ± 9.9 ^a^	102.6 ± 0.1 ^a^	115.6 ± 2.5 ^a^
Ethyl hydrogen succinate (μg/L)	183.0 ± 3.4 ^a^	619.7 ± 1.5 ^e^	533.4 ± 4.3 ^b^	739.2 ± 4.9 ^f^	756.3 ± 15.1 ^f^	517.7 ± 12.2 ^b^	525.7 ± 3.8 ^b^	549.9 ± 2.9 ^c^	570.4 ± 8.4 ^d^
Phenethyl acetate (μg/L)	72.3 ± 2.5 ^b^	92.7 ± 1.9 ^e^	114.8 ± 0.1 ^f^	116.5 ± 3.0 ^f^	118.8 ± 1.7 ^f^	42.5 ± 1.9 ^a^	78.0 ± 2.0 ^c^	82.2 ± 3.1 ^c,d^	84.2 ± 0.1 ^d^
Ethyl decanoate (μg/L)	73.4 ± 1.5 ^f^	36.7 ± 1.4 ^c^	52.9 ± 3.0 ^d^	50.8 ± 1.8 ^d^	68.7 ± 0.6 ^e^	23.5 ± 0.3 ^a^	23.0 ± 0.2 ^a^	22.8 ± 1.3 ^a^	28.8 ± 0.7 ^b^
Ethyl vanillate (μg/L)	7.2 ± 0.4 ^a^	16.3 ± 0.8 ^b,c^	17.1 ± 2.1 ^c^	17.9 ± 0.7 ^c^	48.5 ± 1.2 ^d^	16.7 ± 1.3 ^b,c^	15.6 ± 0.5 ^b^	15.6 ± 0.6 ^b^	15.2 ± 0.5 ^b^
Ethyl laurate (μg/L)	34.6 ± 0.7 ^b^	61.1 ± 0.5 ^c^	65.5 ± 0.9 ^d^	72.8 ± 0.7 ^e^	85.6 ± 1.6 ^f^	12.3 ± 1.0 ^a^	11.6 ± 0.3 ^a^	11.6 ± 0.1 ^a^	11.4 ± 0.6 ^a^
Hexyl salicylate (μg/L)	5.9 ± 0.2 ^b^	13.9 ± 0.7 ^e^	15.0 ± 0.5 ^f^	17.6 ± 0.2 ^g^	18.6 ± 4.0 ^h^	4.7 ± 0.1 ^a^	7.0 ± 0.1 ^c^	8.2 ± 0.2 ^d^	8.2 ± 0.1 ^d^
Ethyl myristate (μg/L)	27.5 ± 1.2 ^b^	30.9 ± 0.1 ^c^	33.9 ± 0.2 ^d^	32.7 ± 0.5 ^d^	37.7 ± 1.0 ^e^	23.2 ± 0.4 ^a^	26.2 ± 1.1 ^b^	27.0 ± 1.0 ^b^	27.0 ± 0.3 ^b^
Diisobutyl phthalate (μg/L)	103.4 ± 4.1 ^a^	195.8 ± 10.5 ^c^	233.3 ± 10.7 ^d^	273.8 ± 8.6 ^e^	307.3 ± 7.5 ^f^	107.4 ± 1.3 ^a^	103.4 ± 5.1 ^a^	176.9 ± 7.9 ^b^	183.0 ± 5.7 ^b,c^
Ethyl pentadecanoate (μg/L)	27.8 ± 0.3 ^b^	45.6 ± 1.1 ^d^	32.1 ± 1.3 ^c^	24.5 ± 0.9 ^a^	22.3 ± 2.1 ^a^	30.5 ± 0.1 ^c^	29.7 ± 0.9 ^c^	30.5 ± 0.7 ^c^	25.4 ± 1.4 ^a,b^
Methyl palmitate (μg/L)	28.6 ± 1.5 ^b,c^	102.2 ± 7.8 ^d^	122.7 ± 1.2 ^e^	132.8 ± 1.4 ^f^	140.6 ± 1.6 ^g^	12.0 ± 2.4 ^a^	25.8 ± 0.2 ^b^	33.9 ± 1.1 ^c^	34.2 ± 0.2 ^c^
Dibutyl phthalate (μg/L)	23.4 ± 0.7 ^a^	102.7 ± 1.8 ^d^	152.3 ± 6.8 ^e^	172.0 ± 5.2 ^f^	174.1 ± 4.1 ^f^	25.3 ± 0.3 ^a^	60.4 ± 0.9 ^b^	82.0 ± 4.8 ^c^	85.6 ± 1.1 ^c^
Ethyl palmitate (μg/L)	107.5 ± 0.9 ^a^	139.8 ± 1.0 ^b^	151.9 ± 7.6 ^c^	172.6 ± 6.7 ^d^	178.1 ± 10.2 ^d^	136.9 ± 6.5 ^b^	137.2 ± 3.8 ^b^	171.6 ± 4.8 ^d^	172.4 ± 8.0 ^d^
Ethyl linoleate (μg/L)	5.7 ± 0.2 ^c^	9.3 ± 0.0 ^d^	9.5 ± 0.1 ^d^	9.2 ± 0.2 ^d^	9.3 ± 0.4 ^d^	4.8 ± 0.6 ^b^	4.7 ± 0.2 ^b^	4.3 ± 0.2 ^b^	3.5 ± 0.1 ^a^
Ethyl oleate (μg/L)	21.8 ± 0.2 ^a^	24.2 ± 0.8 ^b,c^	22.0 ± 0.8 ^a,b^	27.6 ± 0.8 ^c^	27.5 ± 1.5 ^c^	21.2 ± 0.2 ^a^	20.8 ± 0.9 ^a^	21.3 ± 0.6 ^a^	23.3 ± 0.7 ^b^
Ethyl stearate (μg/L)	47.4 ± 1.9 ^a^	58.5 ± 1.3 ^c^	60.0 ± 1.6 ^c^	58.1 ± 0.3 ^c^	57.1 ± 1.2 ^c^	54.1 ± 2.5 ^b,c^	55.0 ± 3.4 ^b,c^	58.0 ± 0.2 ^c^	58.1 ± 1.2 ^c^
**∑Volatile phenols (mg/L)**	1.76 ± 0.07 ^b^	1.65 ± 0.07 ^b^	1.94 ± 0.04 ^c^	2.19 ± 0.03 ^d^	2.37 ± 0.06 ^e^	1.16 ± 0.02 ^a^	1.58 ± 0.04 ^b^	2.02 ± 0.06 ^c^	2.09 ± 0.07 ^c,d^
4-ethylphenol (μg/L)	624.8 ± 25.7 ^g^	419.6 ± 1.1 ^d^	490.7 ± 10.1 ^e^	568.5 ± 10.7 ^f^	647.2 ± 29.1 ^g^	307.2 ± 5.3 ^a^	370.8 ± 1.4 ^b^	370.1 ± 2.8 ^b^	397.8 ± 1.0 ^c^
4-ethylguaiacol (μg/L)	20.9 ± 0.3 ^b^	-	-	-	14.3 ± 1.3 ^a^	-	-	-	-
2,4-Di-T-butylphenol (mg/L)	1.11 ± 0.04 ^b^	1.23± 0.07 ^c^	1.45 ± 0.03 ^d^	1.62 ± 0.02 ^e^	1.71 ± 0.03 ^f^	0.85 ± 0.01 ^a^	1.21 ± 0.04 ^c^	1.65 ± 0.06 ^e^	1.69 ± 0.07 ^e,f^

Different superscript letters in the same row indicate statistical difference by ANOVA, Fisher’s (LSD) test (*p* < 0.05). “-” not detected. Abbreviations: W—initial wine; R—reverse osmosis retentate; 1–2.5 MPa with cooling; 2–3.5 MPa with cooling; 3–4.5 MPa with cooling; 4–5.5 MPa with cooling; 5–2.5 MPa without cooling; 6–3.5 MPa without cooling; 7–4.5 MPa without cooling; 8–5.5 MPa without cooling.

**Table 5 molecules-26-00874-t005:** Aroma compounds identified in Cabernet Sauvignon red wine and nanofiltration retentates at 2.5, 3.5, 4.5 and 5.5 MPa with cooling and without cooling.

Compound	W	N1	N2	N3	N4	N5	N6	N7	N8
**∑Acids (μg/L)**	828.8 ± 29.9 ^h^	372.9 ± 3.0 ^a^	396.3 ± 6.3 ^b^	465.1 ± 6.3 ^e^	738.5 ± 20.8 ^g^	411.6 ± 2.0 ^c^	428.1 ± 6.3 ^d^	509.4 ± 8.0 ^f^	518.6 ± 1.6 ^f^
Acetic acid (μg/L)	682.5 ± 27.6 ^e^	221.0 ± 0.1 ^a^	221.5 ± 3.1 ^a^	229.4 ± 3.4 ^b^	365.8 ± 9.3 ^d^	222.9 ± 0.5 ^a^	223.7 ± 4.6 ^a^	292.5 ± 5.2 ^c^	298.5 ± 0.2 ^c^
Octanoic acid (μg/L)	25.6 ± 0.3 ^a^	40.9 ± 0.4 ^b^	40.5 ± 1.7 ^b^	53.2 ± 2.3 ^d^	60.1 ± 1.2 ^e^	42.1 ± 0.8 ^b^	50.2 ± 0.2 ^c^	51.6 ± 0.6 ^c,d^	53.1 ± 0.4 ^d^
Nonanoic acid (μg/L)	-	-	-	-	-	-	-	-	-
Decanoic acid (μg/L)	65.4 ± 0.8 ^a^	57.7 ± 1.5 ^a^	67.4 ± 0.2 ^b^	71.8 ± 0.1 ^b^	176.5 ± 6.2 ^e^	77.4 ± 0.1 ^c^	81.7 ± 0.4 ^c,d^	82.0 ± 1.1 ^d^	83.6 ± 0.3 ^d^
Lauric acid (μg/L)	16.5 ± 0.1 ^a^	20.2 ± 0.8 ^b^	23.5 ± 0.4 ^c^	27.1 ± 0.1 ^d^	30.2 ± 0.6 ^f^	28.7 ± 0.3 ^e^	30.6 ± 0.7 ^f^	36.6 ± 0.9 ^g^	37.0 ± 0.4 ^g^
Myristic acid (μg/L)	24.8 ± 0.5 ^a^	27.1 ± 0.1 ^b^	37.4 ± 0.8 ^d^	76.1 ± 0.3 ^f^	96.9 ± 3.3 ^g^	35.3 ± 0.2 ^c^	36.6 ± 0.3 ^d^	39.0 ± 0.1 ^e^	38.8 ± 0.2 ^e^
Palmitic acid (μg/L)	14.0 ± 0.6 ^e^	6.0 ± 0.1 ^b^	6.0 ± 0.1 ^b^	7.5 ± 0.1 ^c^	9.0 ± 0.2 ^d^	5.2 ± 0.1 ^a^	5.3 ± 0.1 ^a^	7.7 ± 0.1 ^c^	7.6 ± 0.1 ^c^
**∑Alcohols (mg/L)**	7.29 ± 0.21 ^b^	9.19 ± 0.33 ^c,d^	9.97 ± 0.21 ^d^	11.20 ± 0.23 ^e^	12.26 ± 0.29 ^f^	5.38 ± 0.17 ^a^	7.30 ± 0.14 ^b^	7.66 ± 0.21 ^b^	8.84 ± 0.17 ^c^
Isoamyl alcohol (mg/L)	3.98 ± 0.12 ^b^	5.82 ± 0.09 ^e^	5.95 ± 0.08 ^e^	6.65 ± 0.11 ^f^	7.16 ± 0.11 ^g^	3.00 ± 0.03 ^a^	4.59 ± 0.03 ^d^	4.26 ± 0.07 ^c^	4.69 ± 0.09 ^d^
1-butanol (mg/L)	1.06 ± 0.06 ^b^	0.91 ± 0.09 ^b^	1.43 ± 0.02 ^d^	1.52 ± 0.02 ^e^	1.75 ± 0.08 ^f^	0.57 ± 0.01 ^a^	0.62 ± 0.01 ^a^	1.06 ± 0.07 ^b^	1.28 ± 0.02 ^c^
2,3-butanediol (mg/L)	0.30 ± 0.02 ^a^	0.45 ± 0.01 ^b^	0.54 ± 0.02 ^c^	0.70 ± 0.07 ^d^	0.96 ± 0.02 ^e^	0.38 ± 0.07 ^a^	0.46 ± 0.01 ^b^	0.48 ± 0.01 ^b^	0.55 ± 0.01 ^c^
1-hexanol (μg/L)	42.4 ± 0.7 ^c^	44.8 ± 0.1 ^d^	44.2 ± 0.3 ^d^	44.3 ± 0.3 ^d^	47.0 ± 1.1 ^e^	13.7 ± 0.9 ^a^	15.1 ± 0.1 ^b^	15.7 ± 0.4 ^b^	15.1 ± 0.1 ^b^
Methionol (μg/L)	20.6 ± 0.8 ^c^	15.3 ± 0.3 ^b^	11.4 ± 0.4 ^a^	-	-	-	-	-	-
Benzyl alcohol (μg/L)	7.2 ± 0.1 ^a^	-	-	-	-	-	-	-	-
1-octanol (μg/L)	14.4 ± 0.3 ^a^	18.5 ± 0.2 ^b^	20.0 ± 0.6 ^c^	22.1 ± 0.1 ^d^	24.4 ± 1.0 ^e^	13.0 ± 1.1 ^a^	17.9 ± 1.0 ^b^	20.1 ± 0.2 ^c^	20.1 ± 0.1 ^c^
2-phenylethanol (mg/L)	1.86 ± 0.01 ^c^	1.92 ± 0.14 ^c^	1.96 ± 0.09 ^c^	2.25 ± 0.03 ^d^	2.30 ± 0.08 ^d^	1.39 ± 0.06 ^a^	1.58 ± 0.09 ^b^	1.81 ± 0.06 ^c^	2.27 ± 0.05 ^d^
Dodecanol (μg/L)	6.0 ± 0.1 ^a^	7.1 ± 0.1 ^b^	11.4 ± 0.1 ^c^	13.4 ± 0.2 ^d^	15.7 ± 0.1 ^e^	12.0 ± 0.1 ^c^	15.4 ± 0.1 ^e^	15.4 ± 0.9 ^e^	15.7 ± 0.1 ^e^
**∑Carbonyl compounds (μg/L)**	20.9 ± 0.8 ^a^	31.0 ± 0.8 ^b^	36.8 ± 1.0 ^c^	40.4 ± 1.5 ^d^	52.0 ± 1.0 ^f^	32.7 ± 0.6 ^b^	41.4 ± 1.1 ^d^	46.9 ± 0.9 ^e^	50.5 ± 0.7 ^f^
4-propylbenzaldehyde (μg/L)	6.7 ± 0.3 ^a^	8.9 ± 0.4 ^b^	13.2 ± 0.1 ^d^	15.1 ± 0.8 ^e^	25.5 ± 0.1 ^g^	11.4 ± 0.3 ^c^	16.0 ± 0.1 ^e^	20.3 ± 0.3 ^f^	20.5 ± 0.1 ^f^
Geranyl acetone (μg/L)	5.3 ± 0.1 ^a^	10.2 ± 0.1 ^c^	10.2 ± 0.7 ^c^	10.5 ± 0.3 ^c^	11.2 ± 0.3 ^d^	8.6 ± 0.1 ^b^	8.6 ± 0.1 ^b^	8.1 ± 0.4 ^b^	8.7 ± 0.1 ^b^
Lily aldehyde (μg/L)	4.4 ± 0.2 ^a^	6.4 ± 0.1 ^b^	7.0 ± 0.1 ^c^	7.7 ± 0.3 ^c^	7.2 ± 0.2 ^c^	6.5 ± 0.1 ^b^	9.8 ± 0.8 ^d^	9.8 ± 0.1 ^d^	10.1 ± 0.3 ^d^
Hexyl cinnamaldehyde (μg/L)	4.5 ± 0.2 ^a^	5.5 ± 0.2 ^b^	6.4 ± 0.1 ^c^	7.1 ± 0.1 ^d^	8.1 ± 0.4 ^e^	6.2 ± 0.1 ^c^	7.0 ± 0.1 ^d^	8.7 ± 0.1 ^e^	11.2 ± 0.2 ^f^
**∑Terpenes (μg/L)**	69.4 ± 3.8 ^a^	100.7 ± 1.7 ^d^	110.8 ± 1.8 ^e^	131.3 ± 1.7 ^f^	134.1 ± 3.2 ^f^	84.9 ± 0.5 ^b^	95.8 ± 0.6 ^c^	102.9 ± 2.1 ^d^	114.4 ± 1.9 ^e^
α-terpinolene (μg/L)	15.9 ± 0.7 ^a^	28.6 ± 0.2 ^d^	29.3 ± 0.6 ^d^	35.8 ± 0.3 ^e^	35.5 ± 1.0 ^e^	21.8 ± 0.1 ^b^	22.2 ± 0.1 ^b^	23.8 ± 0.7 ^c^	24.1 ± 1.3 ^c^
Nerol (μg/L)	5.1 ± 0.1 ^a^	8.3 ± 0.2 ^d^	8.5 ± 0.1 ^d,e^	8.7 ± 0.1 ^e^	9.8 ± 0.1 ^f^	5.3 ± 0.1 ^a^	5.9 ± 0.1 ^b^	7.6 ± 0.2 ^c^	9.8 ± 0.2 ^f^
β-citronellol (μg/L)	24.2 ± 1.0 ^a^	30.7 ± 0.4 ^c^	35.6 ± 0.9 ^d^	44.6 ± 0.3 ^f^	47.5 ± 1.3 ^f^	25.5 ± 0.1 ^a^	26.4 ± 0.2 ^b^	30.5 ± 0.2 ^c^	37.1 ± 0.2 ^e^
β-damascenone (μg/L)	18.9 ± 1.9 ^a^	29.7 ± 0.8 ^c^	32.6 ± 0.1 ^d^	34.0 ± 0.9 ^d^	33.1 ± 0.6 ^d^	26.3 ± 0.1 ^b^	34.1 ± 0.1 ^d^	33.8 ± 0.9 ^d^	36.3 ± 0.1 ^e^
Phenanthrene (μg/L)	5.3 ± 0.1 ^c^	3.4 ± 0.1 ^a^	4.8 ± 0.1 ^b^	8.2 ± 0.1 ^f^	8.2 ± 0.2 ^f^	6.0 ± 0.1 ^d^	7.2 ± 0.1 ^e^	7.2 ± 0.1 ^e^	7.1 ± 0.1 ^e^
**∑Esters (mg/L)**	1.83 ± 0.06 ^b^	1.77 ± 0.07 ^b^	2.03 ± 0.08 ^c^	2.27 ± 0.06 ^d^	2.59 ± 0.11 ^e^	1.59 ± 0.02 ^a^	1.88 ± 0.02 ^b^	2.02 ± 0.09 ^c^	2.28 ± 0.03 ^d^
Ethyl hexanoate (μg/L)	66.7 ± 2.4 ^f^	37.7 ± 0.4 ^c^	40.6 ± 0.5 ^d^	49.1 ± 0.8 ^e^	49.6 ± 0.9 ^e^	21.3 ± 1.5 ^a^	30.4 ± 0.2 ^b^	32.5 ± 0.7 ^b^	37.3 ± 0.1 ^c^
Ethyl 4-hydroxybutanoate (μg/L)	50.2 ± 0.9 ^f^	43.7 ± 1.5 ^e^	32.2 ± 0.8 ^c^	32.9 ± 0.7 ^c^	34.7 ± 0.3 ^d^	23.3 ± 0.1 ^b^	21.3 ± 0.4 ^a^	22.0 ± 0.2 ^a^	22.8 ± 0.4 ^a,b^
Diethyl succinate (mg/L)	0.73 ± 0.02 ^a^	1.01 ± 0.05 ^b^	1.17 ± 0.06 ^c^	1.18 ± 0.04 ^c^	1.44 ± 0.09 ^d^	0.95 ± 0.01 ^b^	1.21 ± 0.01 ^c^	1.22 ± 0.08 ^c^	1.44 ± 0.01 ^d^
Ethyl octanoate (μg/L)	210.7 ± 14.7 ^c^	151.4 ± 4.4 ^a^	155.9 ± 4.4 ^a^	177.2 ± 3.8 ^b^	174.5 ± 4.1 ^b^	151.0 ± 6.2 ^a^	151.8 ± 2.6 ^a^	162.5 ± 1.9 ^a,b^	164.6 ± 7.7 ^a,b^
Ethyl hydrogen succinate (μg/L)	183.0 ± 3.4 ^c^	158.4 ± 3.3 ^b^	200.3 ± 2.4 ^d^	334.6 ± 8.6 ^e^	361.3 ± 7.6 ^e^	114.3 ± 2.0 ^a^	112.7 ± 5.2 ^a^	204.3 ± 0.4 ^d^	211.3 ± 3.3 ^d^
Phenethyl acetate (μg/L)	72.3 ± 2.5 ^c^	52.3 ± 2.2 ^a^	73.9 ± 1.2 ^c^	83.9 ± 0.8 ^d^	91.5 ± 2.7 ^e^	51.5 ± 0.9 ^a^	63.9 ± 0.3 ^b^	70.8 ± 1.1 ^c^	83.8 ± 0.2 ^d^
Ethyl decanoate (μg/L)	73.4 ± 1.5 ^g^	36.9 ± 0.3 ^d^	36.9 ± 1.1 ^d^	40.1 ± 0.6 ^e^	46.7 ± 0.9 ^f^	27.0 ± 0.1 ^a^	30.2 ± 0.2 ^b^	32.7 ± 0.1 ^c^	33.6 ± 0.4 ^c^
Ethyl vanillate (μg/L)	7.2 ± 0.4 ^a^	8.4 ± 0.2 ^b^	9.7 ± 0.1 ^c^	14.9 ± 0.1 ^f^	14.6 ± 0.1 ^f^	9.5 ± 0.1 ^c^	10.7 ± 0.4 ^d^	11.7 ± 0.6 ^d^	13.5 ± 0.4 ^e^
Ethyl laurate (μg/L)	34.6 ± 0.7 ^e^	24.9 ± 1.1 ^b^	28.0 ± 0.7 ^c^	32.2 ± 0.2 ^d^	34.4 ± 0.7 ^e^	17.0 ± 0.1 ^a^	24.5 ± 0.1 ^b^	27.8 ± 0.6 ^c^	28.0 ± 0.1 ^c^
Hexyl salicylate (μg/L)	5.9 ± 0.2 ^a^	8.0 ± 0.1 ^b^	9.8 ± 0.1 ^c^	16.5 ± 0.1 ^f^	18.3 ± 0.6 ^g^	9.2 ± 0.1 ^c^	12.0 ± 0.2 ^d^	13.3 ± 0.1 ^e^	16.2 ± 0.1 ^f^
Ethyl myristate (μg/L)	27.5 ± 1.2 ^d^	16.1 ± 0.4 ^a^	33.2 ± 1.0 ^e^	43.7 ± 0.3 ^f^	44.5 ± 1.0 ^f^	16.6 ± 0.1 ^a^	16.9 ± 0.1 ^a^	19.8 ± 0.4 ^b^	22.5 ± 0.2 ^c^
Diisobutyl phthalate (μg/L)	103.4 ± 4.1 ^f^	21.7 ± 0.4 ^a^	25.0 ± 0.6 ^c^	27.3 ± 0.6 ^d^	29.9 ± 0.8 ^e^	23.2 ± 0.1 ^b^	23.5 ± 0.1 ^b^	25.3 ± 0.4 ^c^	24.4 ± 0.7 ^c^
Ethyl pentadecanoate (μg/L)	27.8 ± 0.3 ^d^	23.9 ± 0.2 ^c^	23.7 ± 0.3 ^c^	23.3 ± 0.2 ^c^	21.9 ± 0.7 ^b^	22.2 ± 0.3 ^b^	22.6 ± 0.1 ^b^	21.0 ± 0.5 ^a,b^	20.4 ± 0.1 ^a^
Methyl palmitate (μg/L)	28.6 ± 1.5 ^d^	21.5 ± 0.2 ^b^	21.9 ± 0.3 ^b^	22.9 ± 0.1 ^c^	23.0 ± 0.1 ^c^	18.6 ± 0.1 ^a^	19.0 ± 0.2 ^a^	18.1 ± 0.3 ^a^	18.6 ± 0.2 ^a^
Dibutyl phthalate (μg/L)	23.4 ± 0.7 ^a^	33.2 ± 0.3 ^d^	41.2 ± 0.3 ^e^	43.2 ± 0.6 ^f^	45.3 ± 1.2 ^g^	28.1 ± 0.1 ^b^	30.7 ± 0.1 ^c^	31.0 ± 0.4 ^c^	31.6 ± 0.6 ^c^
Ethyl palmitate (μg/L)	107.5 ± 0.9 ^e^	88.4 ± 0.8 ^c^	90.8 ± 0.9 ^d^	111.9 ± 1.3 ^f^	113.5 ± 0.7 ^f^	72.5 ± 0.3 ^a^	72.5 ± 0.4 ^a^	84.4 ± 0.2 ^b^	84.6 ± 0.2 ^b^
Ethyl linoleate (μg/L)	5.7 ± 0.2 ^a^	5.1 ± 0.2 ^a^	6.2 ± 0.4 ^b^	6.8 ± 0.1 ^b^	7.6 ± 0.2 ^c^	5.5 ± 0.1 ^a^	5.6 ± 0.1 ^a^	5.3 ± 0.1 ^a^	5.1 ± 0.4 ^a^
Ethyl oleate (μg/L)	21.8 ± 0.2 ^f^	14.4 ± 0.1 ^d^	13.0 ± 0.7 ^c^	14.8 ± 0.3 ^d^	15.7 ± 0.3 ^e^	13.7 ± 0.1 ^c^	11.7 ± 0.1 ^b^	10.1 ± 0.6 ^a^	10.3 ± 0.1 ^a^
Ethyl stearate (μg/L)	47.4 ± 1.9 ^e^	15.3 ± 0.3 ^c^	15.4 ± 0.7 ^c^	17.5 ± 0.1 ^d^	18.1 ± 0.7 ^d^	11.5 ± 0.5 ^b^	11.5 ± 0.1 ^b^	9.9 ± 0.2 ^a,b^	9.5 ± 0.1 ^a^
**∑Volatile phenols (mg/L)**	1.76 ± 0.07 ^c,d^	1.57 ± 0.02 ^c^	1.64 ± 0.02 ^c^	1.85 ± 0.05 ^d^	1.85 ± 0.09 ^d^	1.18 ± 0.03 ^a^	1.19 ± 0.04 ^a^	1.24 ± 0.08 ^a,b^	1.37 ± 0.04 ^b^
4-ethylphenol (μg/L)	624.8 ± 25.7 ^d^	193.9 ± 2.8 ^b^	193.8 ± 1.5 ^b^	224.5 ± 3.8 ^c^	225.3 ± 6.5 ^c^	182.9 ± 4.6 ^a^	186.1 ± 0.6 ^a^	189.7 ± 3.5 ^a^	196.7 ± 3.7 ^b^
4-ethylguaiacol (μg/L)	20.9 ± 0.3 ^b^	-	-	-	12.9 ± 0.1 ^a^	-	-	-	-
2,4-Di-T-butylphenol (mg/L)	1.11 ± 0.04 ^b^	1.38 ± 0.02 ^c^	1.45 ± 0.02 ^c^	1.63 ± 0.05 ^d^	1.62 ± 0.08 ^d^	1.00 ± 0.03 ^a^	1.00 ± 0.04 ^a^	1.05 ± 0.08 ^a,b^	1.17 ± 0.04 ^b^

Different superscript letters in the same row indicate statistical difference by ANOVA, Fisher’s (LSD) test (*p* < 0.05). “-” not detected. Abbreviations: W—initial wine; N—nanofiltration retentate; 1–2.5 MPa with cooling; 2–3.5 MPa with cooling; 3–4.5 MPa with cooling; 4–5.5 MPa with cooling; 5–2.5 MPa without cooling; 6–3.5 MPa without cooling; 7–4.5 MPa without cooling; 8–5.5 MPa without cooling.

**Table 6 molecules-26-00874-t006:** RO and NF membrane characteristics.

Sample	pH Range	Operating Temperature (°C)	Maximum Pressure (MPa)	Salt Rejection (%)
RO98pHt	2–11	5–60	5.5	≥98 ^1^
NF	3–10	5–50	5.5	≥99 ^2^

^1^ Measured on 2000 ppm NaCl, 1.6 MPa, 25 °C. ^2^ Measured on 2000 ppm MgSO_4_, 0.9 MPa, 25 °C.

**Table 7 molecules-26-00874-t007:** The limit of detection (LOD), limit of quantitation (LOQ) and mass to charge ratio (*m*/*z*) of volatile compounds identified in Cabernet Sauvignon wine, RO and NF retentates.

Compound	LOD	LOQ	*m*/*z*	Compound	LOD	LOQ	*m*/*z*
Acids	µg/L	µg/L		Esters	µg/L	µg/L	
Acetic acid	30.0	100.0	60	Ethyl hexanoate	1.2	4.0	115–144
Octanoic acid	0.9	3.0	101–115	Ethyl 4-hydroxybutanoate	1.2	4.0	132
Nonanoic acid	0.6	2.0	129–158	Diethyl succinate	50.7	169.0	129–174
Decanoic acid	1.2	4.0	129–172	Ethyl octanoate	9.6	32.0	127–172
Lauric acid	0.3	1.0	157–200	Ethyl hydrogen succinate	8.7	29.0	128–146
Myristic acid	0.3	1.0	185–228	Phenethyl acetate	2.7	9.0	104–164
Palmitic acid	0.3	1.0	213–256	Ethyl decanoate	2.7	9.0	155–200
Alcohols				Ethyl vanillate	0.6	2.0	151–196
Isoamyl alcohol	96.0	320.0	70–88	Ethyl laurate	0.9	30.0	183–228
1-butanol	33.6	112.0	56–74	Hexyl salicylate	0.3	1.0	120–222
2,3-butanediol	27.6	92.0	45–90	Ethyl myristate	0.6	2.0	88–256
1-hexanol	1.8	6.0	84–102	Diisobutyl phthalate	3.3	11.0	149–278
Methionol	0.6	2.0	88–106	Ethyl pentadecanoate	1.8	6.0	225–270
Benzyl alcohol	0.3	1.0	79–108	Methyl palmitate	1.5	5.0	239–270
1-octanol	0.3	1.0	112–130	Dibutyl phthalate	1.2	4.0	149–278
2-phenylethanol	87.3	291.0	91–122	Ethyl palmitate	5.4	18.0	239–284
Dodecanol	0.3	1.0	168–186	Ethyl linoleate	0.3	1.0	263–308
Carbonyl compounds				Ethyl oleate	0.6	2.0	264–310
4-propylbenzaldehyde	0.3	1.0	119–148	Ethyl stearate	1.5	5.0	267–312
Geranyl acetone	0.3	1.0	176–194	Volatile phenols			
Lily aldehyde	0.3	1.0	189–204	4-ethylphenol	1.8	6.0	107–122
Hexyl cinnamaldehyde	0.3	1.0	129–216	4-ethylguaiacol	0.9	3.0	137–152
Terpenes				2,4-Di-T-butylphenol	54.3	181.0	191–206
α-terpinolene	0.3	1.0	121–136				
Nerol	0.3	1.0	139–154				
β-citronellol	0.6	2.0	138–156				
β-damascenone	0.3	1.0	175–190				
Phenanthrene	0.3	1.0	178				

## Data Availability

Not available.

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
