# Peer review of "Aroma Profile and Chemical Composition of Reverse Osmosis and Nanofiltration Concentrates of Red Wine Cabernet Sauvignon"

_molecules, 2021, doi:10.3390/molecules26040874_

Round 1

Reviewer 1 Report

The aim of the reviewed manuscript is to obtain red wine concentrates with an enriched and corrected content of aromatic compounds as a way to improve the quality of the finished product that meets the expected and high standards. The authors of this manuscript chose Cabernet Sauvignon red wine for the research, which was processed by reverse osmosis and nanofiltration under various pressure and temperature conditions, using several composite polyamide membranes type RO98pHt M20 and NF M20. During the research, the influence of the applied separation method on the change of the composition of volatile aromatic compounds analyzed with the SPME GC-MS technique was assessed. The changes in the content of reducing sugars, acids, sulfur dioxide and ethanol in the initial samples of wine and those obtained after the application of the tested concentration methods, determined by classical analytical methods were also compared. To compare the retentates based on the obtained aroma profiles, the principal component analysis was applied.

According to the reviewer, the article was prepared correctly: the validity of the research undertaken was explained; individual research goals and methods of achieving them were defined, the methodology was correctly presented, the results documenting the achievement of the intended goals were also adequately supported by tables and figures. Therefore, I believe that this manuscript deserves to be published in its present version.

Author Response

Dear Reviewer,

thank you very much for your comments. We are happy that you are satisfied with our manuscript.

Best regards!

Reviewer 2 Report

The paper is interesting and made  according to modern trends in analytical chemistry, but I have some remarks.

bar for pressure and M for concentration aren't in accordance with SI system

In tables - what is a, b .....f

There is the lack of method validation (LOD, LOQ)

Author Response

Dear Reviewer,

thank you for your comments. We have made changes according to your suggestions and you can find the point-by-point response in the attachment.

Best regards!

Reviewer 3 Report

The authors prepared red wine concentrates with enriched achemical composition. They concentrated Cabernet Sauvignon red 
wine varietyby reverse osmosis (RO) and nanofiltration (NF) processes under after evaluating different operating conditions.

The experimental part is well written and discussed. There are some minor syntax comments that could be fixed during revision.

line 17, remove "content"

line 62, remove "up"

line 66, instead of "used" write "apply"

line 76, remove "in order"

Figure 2 is informative, but low resolution, please improve 

line 232 "significant loss of most carbonyl compounds and terpenes was observed with the regime without cooling", can you expalin this?

Begin the last paragraph of the Introduction with the aim of this current study and then describe your previous work 

Rephrase line 302

Author Response

Dear Reviewer,

thank you very much for your comments. We have made changes in the manuscript according to your suggestions. You can find the point-by-point response in the attachment.

Best regards!

Round 2

Reviewer 2 Report

It can be accepted after correction.